# The impacts of conditions and person-organization fit on alliances performance: And the moderating role of intermediary

Hui Sun[1], You-Yu Dai [1,2‡]*, Chaochen Zhang[2], Rok Lee[3], Su-Sung Jeon[4]*, Jin-Hua Chu[5]

1 International Business School, Shandong Jiaotong University, Weihai City, Shandong province, China, 2 Chinese International College, Dhurakij Pundit University, Bangkok City, Thailand, 3 Department of LINC Plus Project Organization, Gyeongsang National University, Jinzhou City, Gyeongsang Namdo, South Korea, 4 Department of Business Administration, Gyeongsang National University, Jinzhou City, Gyeongsang Namdo, South Korea, 5 Department of Development Planning and Disciplinary Construction, Shandong Jiaotong University, Jinan City, Shandong province, China

‡ This author contributed to this work as the joint first author.
* psyyyt@gmail.com (YYD); jss9999@gnu.ac.kr (SSJ)

**Data Availability Statement:** All relevant data are within the paper and its Supporting Information files.

## Abstract

This study expects to provide a reference for the catering industry. The travel industry expands sales channels and turnover tends to choose a strategic alliance with the alliance objects mutually beneficial cooperation to improve their competitiveness. This study examines the effects of alliance conditions and person-organization fit (P-O-fit) on the performance of strategic alliances between travel industries. Furthermore, this study contained the intermediary performance as a moderator to examine the influences of alliance conditions and P-O-fit on the performance of strategic alliances. There were 406 usable questionnaires collected. We verified the hypotheses by the structural equation modeling method. The results suggest that the alliance conditions have positive and significant direct effects on the performance of strategic alliances. Moreover, the P-O-fit also has positive and significant effects on the performance of strategic alliances. Furthermore, the intermediary performance has substantial moderating effects on the influences of P-O-fit on the performance of strategic alliances. The conclusion provides a theoretical and practical basis between performance and the travel industry.

## 1. Introduction

Due to the rapid changes in globalization thought and technology integration, the strategic alliance has become one of the world's most popular business cooperation choices [1]. Among the many strategic options for enterprises, strategic alliances have access to regional markets, overcome barriers to entry, gain market influence, maintain market stability, obtain technological products or new technologies, associate resources, reduce uncertainty, and share the benefits of the risks of research and development plans, the acceleration of new market entry, the combination of complementary assets to increase the source of income, etc. [2–4]. Therefore, more and more enterprises participate in strategic alliances, hoping to increase the scope of operations and gain a comparative advantage in the industry, thereby reducing costs,

**Funding:** You-Yu Dai Grant number YYX20211204 Special project for construction theory of applied undergraduate university in Shandong Jiaotong University http://fzghc.sdjtu.edu.cn/ The funders had no role in study design, data collection and analysis, decision to publish, or preparation of the manuscript. Jinhua Chu Grant number 20BGL219 The National Social Science Fund of China http://www.nopss.gov.cn/ The funders had no role in study design, data collection and analysis, decision to publish, or preparation of the manuscript.

**Competing interests:** The authors have declared that no competing interests exist.

increasing operating performance or firm performance, resource and knowledge sharing, and improving profitability [5–8].

However, alliances are often accompanied by a higher rate of alliance failures. Research on the problems of alliance performance has always been a difficult and hot point in alliance-related studies [9–12] both pointed out that good alliance conditions characteristics (aggregation intensity, conflict between partners, interdependence) produced by cooperation contribute to alliance performance. Among them, sufficient resources and operational capabilities brought about by partnership, that is, the increase in the intensity of aggregation, and the interdependence between partners, contribute to the improvement of alliance performance. If the alliance relationship is in a state of conflict, it will weaken the alliance's performance [13].

Chen et al. [14] showed that the higher the employee's personal-organization fit (P-O-fit), the higher the job performance. Wombacher and Felfe [15] also showed that when employees' values and beliefs are consistent with the organization, employees are willing to contribute to the organization. Because employees recognize that they are one with the organization, they identify with it and work for it, affecting job performance [16]. It is consistent with the conclusion of Chen et al. [14] that personal-organization fit is positively correlated with work performance. Therefore, the personal-organization fit will indeed affect the alliance's performance. The higher the intermediary performance, the stronger the export trade performance [17, 18], which explains that intermediary performance has a positive impact on alliance performance.

Nowadays, various industries such as airlines, travel services, telecommunications, software, hardware and educational services, and automotive sectors require alliances as key growth strategies to improve performance [19–21]. [22] divided strategic partnerships in the travel industry into industry alliances and different industry alliances. The strategic partnership between industry and collective structure can enable participating companies to reduce costs, share risks, and achieve economic performance [23, 24]. Therefore, there is an increasing trend of strategic alliances among travel-related industries. In the cross-industry coalition, for example, the travel industry cooperates with the credit card issuing bank to issue co-branded cards. Such as, the Commonwealth Bank issues a "Federal Travel Card". In addition to the travel fund feedback, it cooperates with the travel industry to promote itineraries and cooperate with website advertising. It is an example of the application of cross-industry alliances in the travel industry. Therefore, how to improve the alliance's performance and make profits is a topic of concern to the travel industry.

Meanwhile, in the past, most studies only considered the direct impact of intermediary performance on work performance. Still, they never felt the research on intermediary performance's effect as a moderating variable. In this research, we verified the moderating role of intermediary performance. Further, we consider the P-O fit factor to affect alliance performance.

In summary, forming alliances combines partners' resource advantages to improve performance and assist competition [25]. Therefore, choosing the right partner has become an important topic. In addition, how different alliance conditions caused by cooperation with partners affect the performance of travel industry strategic alliances is also a topic worthy of discussion. Given this, this study expects that when choosing alliance partners, the travel industry should carefully evaluate the fit between them and their enterprises to increase the chances of success in strategic alliances.

## 2. Literature review and hypotheses development

Alliance conditions are the characteristics of an alliance at any given moment in the coalition's life [26]. When an enterprise forms a strategic alliance based on the ability of alliance partners

to make up for their shortcomings in terms of resources, achieving strategic goals, etc., it will inevitably lead to different alliance conditions with partners in the alliance [27]. Till now, there is no universal standard for the characteristics of alliance conditions [11]. Described the characteristics of alliance conditions in terms of collective strength, conflicts between partners, and interdependence [28, 29]. Also adopted this feature. Hence, this article still uses these classification features.

Chatman [30] studied socialization's impact on value alignment. Firstly, new employees who fit in with the company's values are better suited to the organization. Secondly, those who accept a more rigorous socialization process are better suited to the company's values. Finally, recruits' values are more compatible with the importance of the company. These recruits are more satisfied with the company and tend to stay longer. After the socialization process, the degree of conformity at different levels affects the outcomes of individual behaviors, and there are two consequences. One is the individual-level work attitude, which includes job satisfaction, organizational commitment, turnover intention, work pressure, organizational citizenship behavior, work performance, etc. Another is the executive level. It means that employees with a high degree of fit have a more positive attitude. In this paper, the "individual" in the person-organization fit is substituted into a single travel agency organization, and the "organization" is other corporate entities related to the travel agency. Then the relationship between the person-organization fit and alliance performance is studied accordingly.

## 2.1. The impact of alliance conditions characteristics on alliance performance

The alliance strategy is one solution to face the speed of competition in the business world or business as cooperative strategies in the form of partnerships [31]. One of the required elements for a successful alliance is the alliance condition that drives to promote the alliance's performance. However, in the case of alliance conditions' negative views such as partner opportunism, task complexity, etc., an association would face a challenge [10, 32].

Accordingly, the positive factors of the collective strength, overall resources, or competitiveness of the enterprises in the alliance contribute to the alliance's performance [25]. In other words, the alliance strategy can achieve goals through the power of joint partners. The resources of a single enterprise will not always be beneficial with the help of the strategic alliance's collective partners. It can also ensure that the alliance can achieve performance. In related research, the collective strength of the partnership is positively correlated with alliance performance [33, 34].

Chen [35] divided conflicts into interest conflicts and operational conflicts. Neither of these two conflicts is conducive to alliance performance. Yang et al. [36] believed destructive conflicts could reduce alliance performance. When companies have different interests and competitions are formed, the motivation and incentives to work together will decrease, and the team's job satisfaction will also decrease [37]. Also believed that conflict between partners is negatively correlated with alliance performance. In strategic alliances, the party with high dependence will usually be regarded as having no arrangement. It will be devalued by the other party's partners, believing it has not contributed to the alliance's performance. Malagueño et al. [38] proposed that highly balanced dependence positively affects alliance performance [39, 40]. Also pointed out that the interdependence between partners positively correlates with alliance performance.

Thus, here is the following hypothesis:

**Hypothesis 1 (H1).** The characteristics of alliance conditions positively affect alliance performance.

## 2.2 The impact of person-organizational fit on alliance performance

Employees and their attitudes and behaviors towards their organization are highly anticipated among both organizations and scholars [41]. In other words, congruence of members and their attitudes and behaviors towards their organization as values and goals, P-O fit in the view of the supplementary fit, drives to contribute to the organization and active cooperation among members, resulting in increasing the organizational performance [42, 43].

"Person-organization fit" is defined as congruence between an individual and their organization in terms of such dimensions as values and goals [44]. The early research literature on person-organization fit is based on the Attraction-Selection-Attrition (ASA) analysis framework [45]. The attraction-selection-retention structure describes when the background characteristics of individuals and organizations are similar. They will attract each other so that employees can enter the organization and place them in the job positions that are most suitable for performance. The relevant research results on person-organization fit and performance are different. For example [46], examine P-O fit and P-S fit perceptions as related to employee organizational commitment because organizational commitment can be conceived of as a core variable for performance, prosocial behaviors, absenteeism, and turnover. Hamidah et al. [47] mentioned that, with organization fit, teachers would feel comfortable doing their job at school. The comfort will improve the passion for always wanting to improve their work performance. Siyal et al. [44] focused on the positive influence of high-performance human resource practices (HPHRPs) on P-O fit and the full mediation effect of P-O fit on the relationship between HPHRPs and employee outcomes.

Thus, here is the following hypothesis:

**Hypothesis 2 (H2).** Person-organization fit positively affects alliance performance.

## 2.3 The moderating influence of intermediary performance

Travel enterprises must deal with export and import business, like trade enterprises. So, it is known as inbound and outbound travel [48]. Indicated that studies narrowly focus on intermediary performance and its antecedents and consequences. Improving intermediary performance is the key determinant in promoting corporate sustainability. Despite the extensive scholarly research on strategic alliances [49], previous research has not considered *intermediary performance* as a moderator of the impact of various factors influencing alliance performance. This study innovatively adopts intermediary performance from the trade field to examine its role in the travel industry. There is no specific unified definition of intermediary performance. Referring to [50], intermediary performance in this study is the performance of a travel-intermediary-related firm. The previous researchers only test the direct causal relationship of intermediary performance [50]. Measured intermediary performance by acquiring new clients, retaining existing clients, goal achievement, market share, income, and export growth [51]. Empirically tested the causal relationship between economic growth and intermediary performance.

Peng and York [52] divided the intermediary performance from the perspective of intermediaries into search costs, negotiation costs, and monitoring or implementation costs. The search cost generally involves acquiring knowledge through market research and planning in advance, and obtaining such knowledge without external help can be expensive and time-consuming [53]. High search costs not only prevent many importers and exporters of the tourism industry from expanding internationally, but they may also lead to vague searches in the past, thereby increasing the possibility of bad debts in the cost and affecting exports [54]. At this time, travel agencies can help, by providing experience in understanding foreign markets

and export processes, so that the alliance's tourism industry players can be familiar with international marketing strategies.

Negotiation costs include the direct costs of travel and personnel and the ex-ante costs of dealing with potential dangers with unfamiliar foreign customers [55]. Tourism imports and export companies may lack experience in foreign markets in this negotiation. Because of their lack of cultural understanding, the complexity of negotiation norms is derived. Because of their expertise, tourism agencies will assist the alliance's tourism industry players in reducing negotiation costs.

There are many studies work about moderators in strategic alliances of travel industry and others. For instance, Tsaur and Wang [56] found that competitive intensity moderates the relationship between strategic alliance and performance. Moghaddam et al. [57] observed SEFs benefit from strategic alliances more if they engage in a moderate number of alliances rather than being overwhelmed with a great number of alliances. Wang et al. [58] indicated structural capital positively and significantly moderates the mediating effect on the relationship between complementary capabilities and supplier performance. Overall, travel intermediaries can provide professional product choices for travel importers and exporters. Transaction cost theory believes that because the distribution of complex products is more likely to require significant asset-specific investments, such as professional sales force training and after-sales services, these products will require more channel integration. Once the contract is signed, the parties concerned are about the contract obligations and implementation of the subsequent supervision. The tourism agency can help the alliance's tourism industry reduce additional costs.

Therefore, the higher the intermediary performance, the better the relationships between alliance conditions, P-O fit, and alliance performance.

Thus, here are the following hypotheses:

**Hypothesis 3 (H3).** The high intermediary performance will moderate the positive relationship between alliance conditions and alliance performance.

**Hypothesis 4 (H4).** The high intermediary performance will moderate the positive relationship between person-organization fit and alliance performance.

## 2.4 Research model

Based on previous discussions in related theories and literature, this study intends to explore the impact of alliance conditions characteristics and person-organization fit on alliance performance and adopts intermediary performance as moderating variable. As shown in Fig 1, the research framework of this paper assumes the attributes of strategic alliance conditions (collective strength, conflict between partners, and interdependence) and person-organization fit as independent variables, alliance performance as a dependent variable, and intermediary performance as a moderating variable.

This conceptual framework is clear that the travel industry not only values the characteristics of alliance conditions but also focuses on the influence of person-organization fit on alliance performance when establishing alliance objects. Based on the investigation and statistical analysis, the rationality and universality of the conceptual model are verified. The moderating role of intermediary performance between alliance conditions, person-organization fit, and alliance performance are also discussed.

## 3. Research design

This survey was approved by the institutional ethics committee of International Business School at Shandong Jiaotong University. Most important, this study was analyzed

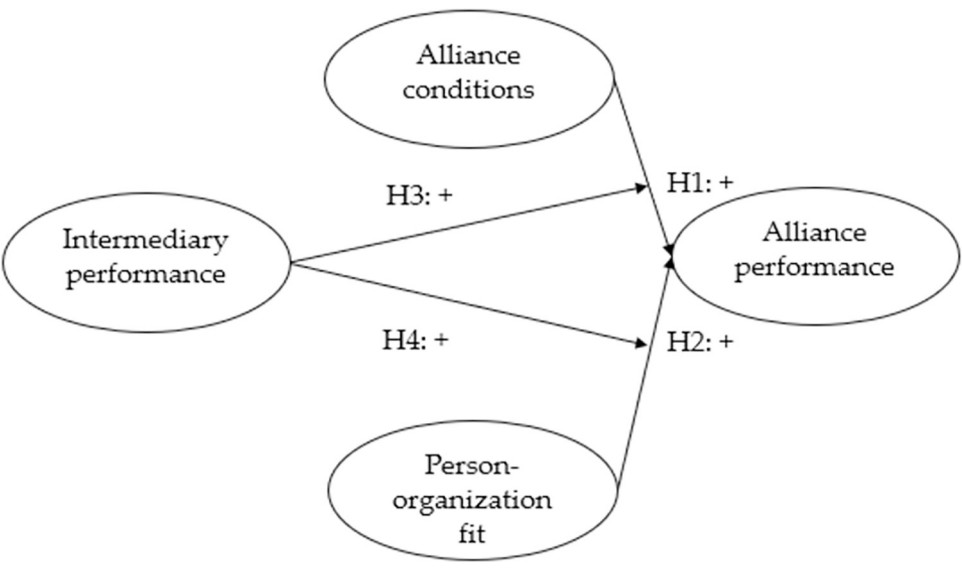

**Fig 1. Research model.**

anonymously. Participants were sufficient understanding the research purpose and finish this questionnaire by self-report. We had provided a paragraph of guidelines to ensure every consent was informed, as following,

> Thank you for taking the time to take part in the questionnaire survey. This is a study on the business model of enterprises. Your valuable opinions will be of great help to this study. The personal data is only for academic research purposes and is not used for other purposes or disclosed to the public. Please fill in the information at ease. Thank you for your cooperation and assistance. Wish you good health and everything goes well.

## 3.1 Questionnaire

This study took the sample from personnel in travel-related industries with experience in a strategic alliance. The travel-related industries include such as travel agencies, airline companies, hotels, and hospitality stores. To effectively spread the questionnaire, the research process is conducted through an online survey. This study used the snowball sampling method to distribute the questionnaire. The main survey channels were via WJX (a Chinese-language website for creating a survey), QQ (an instant messaging software service developed in China), WeChat (a Chinese social media app), Weibo (a Chinese microblogging website), and e-mail services [59]. In this way, the researchers would interact with the invitees on time, explain the filing requirements of the questionnaire for the first time, and be able to answer various questions of the interviewees in time.

The questionnaires were divided into alliance conditions characteristics (collective strength, conflict between partners, and interdependence), person-organization fit (supplementary fit, complementary fit), alliance performance, and intermediary performance (search cost, negotiation cost, and monitoring/implementation cost). The alliance conditions scale was adopted from Lee's [12] 8 items; the alliance performance scale was adopted from Lee's [12] 3 items. This study modified the person-organization fit scale from Lo and Hsieh's [60] 18 items. The intermediary performance scale was adopted from Peng and York's [52] 10 items. This study also used Likert's 5-point scale to measure from 1 (strongly disagree) to 5 (strongly agree).

## 3.2 Data

This study was conducted from December 30 to January 15, 2021, and formal questionnaires were distributed through the incentive online questionnaire survey. A total of 480 question-naires were distributed, and 455 questionnaires were recovered, with a recovery rate of 92%. The recovered samples were screened, and the invalid questionnaires (such as incomplete questionnaires) were deleted. A total of 406 questionnaires were effectively recovered, with an effective recovery rate of 89%. The number of valid questionnaires meets the criteria of more than 10 times the number of questionnaire items [59].

The demographic analysis (Table 1) showed that most of the gender is female (52.5%). The majority of age is 31 to 40 years old (33.7%). The majority of educational background is under-graduate (62.3%), and the majority of marital status is married (61.13%). The current working time in the company is mainly 1 to 6 years (57.2%). The position is mostly below middle man-agers (94.3%). In the industry, other travel-related sectors accounted for the majority (27.8%). Among all respondents' monthly income, the majority (89%) were below 10,000 RMB.

In this study, anonymous responses were set to control for possible common method bias, and the common method bias was tested using the Harman single factor method [61]. The results indicated that no serious common method bias existed in this study.

Before conducting a confirmatory analysis, we must assess whether the data is in line with how normal it is. A total of 39 items are composed of 4 dimensions. The skewness of each item is between -0.684 and 0.106, and its absolute value is less than the standard of 3.0 [62]. The kurtosis of each item is between -0.909 and 0.862, and its total value is less than the standard of 10.0 [62]. These results mean that the questionnaire items in this study follow a normal distri-bution and can be estimated in the structural equation model by the most probable approxi-mation method.

**Table 1. Demographic statistics.**

| Personal attributes | Classification | Number of samples | Proportion (%) | Personal attributes | Classification | Number of samples | Proportion (%) |
|---|---|---|---|---|---|---|---|
| gender | male | 193 | 47.5 | Education level | High school | 30 | 7.4 |
| | female | 213 | 52.5 | | College | 63 | 15.5 |
| age | 18–25 | 83 | 20.4 | | University | 253 | 62.3 |
| | 26–30 | 123 | 30.3 | | Graduate | 60 | 14.8 |
| | 31–40 | 137 | 33.7 | position | Intern | 40 | 9.9 |
| | 41–50 | 39 | 9.6 | | Ordinary employees | 122 | 30.0 |
| | 51 and above | 24 | 5.9 | | Grassroots managers | 104 | 25.6 |
| Marriage status | married | 248 | 61.1 | | Middle managers | 117 | 28.8 |
| | Unmarried and others | 158 | 38.9 | | Top managers | 23 | 5.7 |
| Work years | 1–3 | 120 | 29.6 | Travel-related industries | Catering | 66 | 16.3 |
| | 4–6 | 112 | 27.6 | | accommodation | 17 | 4.2 |
| | 7–9 | 72 | 17.7 | | Transportation | 68 | 16.7 |
| | 10 and above | 102 | 25.1 | | Travel industry (such travel agency) | 37 | 9.1 |
| Monthly income (RMB) | Less than 6000 | 159 | 39.2 | | Cultural and entertainment industry (such as scenic spots, museum, park) | 78 | 19.2 |
| | 6001–8000 | 119 | 29.3 | | Tourism and shopping | 27 | 6.7 |
| | 8001–10000 | 83 | 20.4 | | Other travel-related industries | 113 | 27.8 |
| | 10001 and above | 45 | 11.0 | | | | |

In this study, the Cronbach's α value of each factor is more significant than 0.75, which means the questionnaire reliability is good. In addition, the KMO value of an overall sample is 0.96, and the Sig value of Bartlett's sphericity test is 0.000. Therefore, there is a correlation between the variables, indicating that the validity is suitable for factor analysis.

## 4. Results

Before conducting confirmatory analysis, this study first evaluates whether the data conforms to the multivariate normality. A total of 39 questions are composed of four facets. The skewness of each question is between -0.684 and 0.106, and the absolute value is less than the standard of 3.0 [63, 64]. The kurtosis of each question is between -0.909 and 0.862, and its absolute value is less than the standard of 10.0 [63, 64]. It indicates that the questions in this study are normally distributed and can be estimated by the maximum likelihood method. Therefore, this study uses the maximum likelihood method to estimate the structural equation model.

The KMO value of the whole sample is 0.96, which indicates that it is very suitable for factor analysis [59, 63]. The sig value of Bartlett's sphericity test is 0.000, which is less than the significance level of 0.05. Therefore, there is a correlation between variables, which is suitable for factor analysis. According to the exploratory factor analysis results of the alliance condition scale, three factors were obtained in this study, which were named aggregation intensity, conflict, and interdependence among partners, and could explain 76.52% of the variables. The KMO is 0.85. The exploratory factor analysis results of the individual organization fit scale show that there are two dimensions, namely, complementary fit and complementary fit, which can explain 71.49% of the variables. KMO was 0.96. The exploratory factor analysis results of the intermediary performance scale have three dimensions, namely, search cost, and negotiation comonitoring / implementationntation cost, which can explain 81.77% of the variables. KMO was 0.96. The above results indicate that the alliance status, individual organization fit, and intermediary performance scale of this study have enough internal correlation in the sampling data, and can reasonably conduct exploratory factor analysis.

### 4.1 Confirmatory factor analysis

After previous exploratory factor analysis, this research used confirmatory factor analysis (CFA) to realize the factor structure of alliance conditions, person-organization fit, alliance performance, and intermediary performance. The SMC values of the alliance conditions scale roughly fall between 0.54 and 0.75, the SMC values of the personal-organization fit scale are between 0.59 and 0.86, the SMC value of the alliance performance scale is between 0.76 and 0.83, and the SMC value of intermediary performance scale is between 0.53 and 0.74 (see Table 2). These SMC values show that each measurement item has good explanatory power. In addition, the t value between each item is more significant than 1.96, indicating that the items have reached a considerable level.

The composite reliability (CR) of potential disguised phases is the reliability of all measured disguised steps, representing the strength of internal consistency. The higher the value, the higher the character of the measurement variable [59, 65]. All suggested that CR should be greater than 0.7. The CR values of all constructs (0.85 to 0.97) in this study align with the standard (see Table 2), showing high internal consistency.

Based on the above analysis results, the standardization factor loading (SFL), construction reliability (CR), and average variation extraction (AVE) indicators of this study all reach the recommended value of scholars [59, 65] which indicate that this research questionnaire has good reliability and convergent validity.

**Table 2. CFA results.**

| Factors/ items | SFL | SE. | t-value | SMC | CR | AVE |
|---|---|---|---|---|---|---|
| **Alliance conditions.** | | | | | **0.87** | **0.50** |
| 1. Owing to our cooperation, we have plenty of resources. | 0.75 | 0.05 | 14.82 | 0.74 | | |
| 2. Owing to our cooperation, we have the operating capacity. | 0.67 | 0.05 | 12.91 | 0.66 | | |
| 3. I think state partnerships often strain the alliance. | 0.62 | 0.07 | 11.85 | 0.61 | | |
| 4. I think the executive's alliance work often had incompatible situations. | 0.64 | 0.07 | 12.27 | 0.63 | | |
| 5. How to work for an alliance between the members often conflicts. | 0.55 | 0.07 | 10.49 | 0.54 | | |
| 6. Partners among dependent on each other. | 0.71 | 0.07 | 13.70 | 0.70 | | |
| 7. Partners are challenging to replace each other. | 0.75 | 0.06 | 14.60 | 0.75 | | |
| 8. The partners feel that the cost of each loss is very high. | 0.70 | 0.06 | 13.37 | 0.69 | | |
| **P-O fit** | | | | | **0.97** | **0.64** |
| 1. My organization and our coalition partners will play a team. | 0.84 | 0.05 | 20.51 | 0.84 | | |
| 2. In the team, we respect and trust employees. | 0.84 | 0.05 | 20.62 | 0.84 | | |
| 3. We emphasize task-oriented. | 0.67 | 0.05 | 14.90 | 0.66 | | |
| 4. Emphasis on institutional regulations. | 0.59 | 0.05 | 12.83 | 0.59 | | |
| 5. Attention to human resources development. | 0.84 | 0.05 | 20.34 | 0.83 | | |
| 6. Emphasize management objectives. | 0.86 | 0.05 | 21.11 | 0.85 | | |
| 7. Emphasis on the harmony of personal goals with organizational goals. | 0.74 | 0.05 | 17.15 | 0.74 | | |
| 8. Importance to the work of the target issue. | 0.86 | 0.05 | 21.17 | 0.85 | | |
| 9. Get a sense of accomplishment and self-growth from the cooperative work. | 0.81 | 0.05 | 19.37 | 0.80 | | |
| 10. Achieve organizational vision in cooperation work. | 0.82 | 0.05 | 19.76 | 0.81 | | |
| 11. It provides good opportunities for cooperation. | 0.84 | 0.05 | 20.43 | 0.83 | | |
| 12. We can get a good organizational performance. | 0.76 | 0.05 | 17.78 | 0.76 | | |
| 13. To get the relationship and communication between organizations. | 0.81 | 0.05 | 19.06 | 0.81 | | |
| 14. We have a good working environment. | 0.80 | 0.05 | 19.18 | 0.80 | | |
| 15. Our work time is based on the alliance's needs. | 0.78 | 0.05 | 18.36 | 0.77 | | |
| 16. Pay attention to efficiency. | 0.86 | 0.05 | 21.38 | 0.86 | | |
| 17. My organization is actively involved in the coalition's work. | 0.84 | 0.05 | 20.41 | 0.83 | | |
| 18. We can enrich our affiliate organization with the work required expertise and experience. | 0.82 | 0.05 | 19.73 | 0.81 | | |
| **Alliances performance** | | | | | **0.85** | **0.65** |
| 1. Owing to our cooperation, we get better and cheaper upstream supply (for example, seats and group pricing, etc.) | 0.83 | 0.05 | 15.58 | 0.83 | | |
| 2. Owing to our cooperation with each other, we get higher standards of productivity (For example, the group rate, the number of groups and the ticket number of sheets, etc.) | 0.76 | 0.05 | 16.71 | 0.76 | | |
| 3. Owing to the cooperation with each other, we have reached a higher selling price of the product | 0.82 | 0.06 | 18.32 | 0.82 | | |
| **Intermediary performance** | | | | | **0.95** | **0.67** |
| 1. The firm's overseas connections. | 0.83 | 0.03 | 19.17 | 0.74 | | |
| 2. The firm's industry experience. | 0.86 | 0.03 | 20.54 | 0.73 | | |
| 3. Top-3 managers' industry experience. | 0.86 | 0.03 | 20.29 | 0.70 | | |
| 4. Key people at my firm have better negotiation abilities. | 0.84 | 0.03 | 19.58 | 0.64 | | |
| 5. Key people at my firm handle export negotiations more frequently. | 0.80 | 0.04 | 18.23 | 0.71 | | |
| 6. My firm is more willing to take title to goods. | 0.84 | 0.03 | 19.79 | 0.69 | | |
| 7. My firm has better financial abilities to take title to goods. | 0.83 | 0.04 | 19.40 | 0.64 | | |
| 8. The number of training manufacturers provide to my sales force. | 0.80 | 0.04 | 18.27 | 0.63 | | |
| 9. The amount of training my sales force provides to foreign customers. | 0.79 | 0.04 | 18.05 | 0.53 | | |
| 10. The amount of required after-sales services. | 0.73 | 0.04 | 15.91 | 0.53 | | |

Note: SFL is standard factor loading, SE is the standard error, SMC is square of multi-correlation, CR is composite reliability, and AVE is average variance extracted.

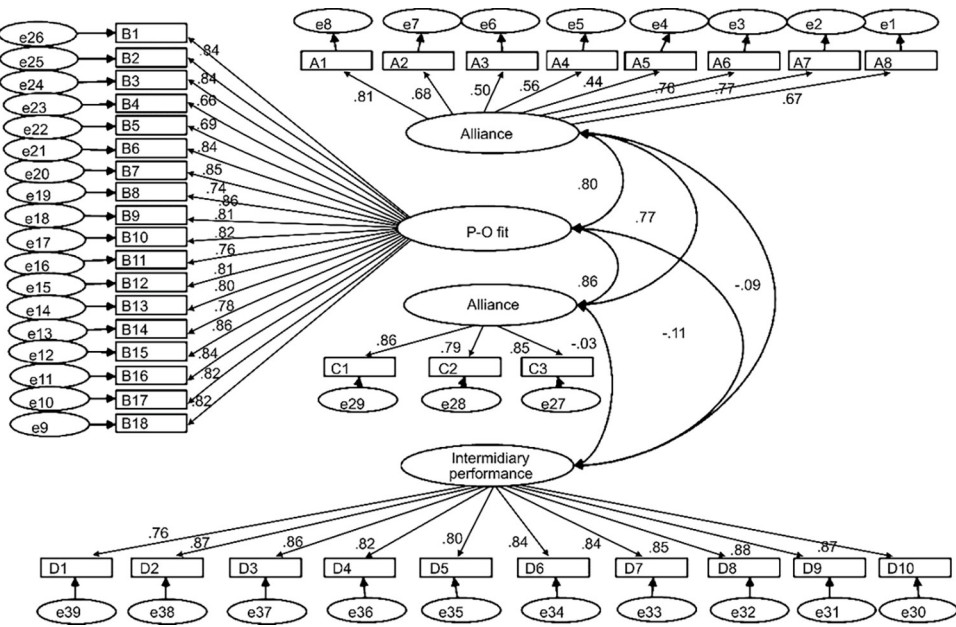

**Fig 2. Diagram of model fit.**

## 4.2 Verification of structural model

In terms of the model fit test, an essential concept of structural equation model evaluation is to evaluate whether the hypothetical model proposed by the researcher is accurate. After each parameter in the structural equation model is successfully estimated, the researchers can judge the fit between the theoretical model and actual observation data through the overall fit evaluation.

**4.2.1. model fits.** For measurement model, GFI, AGFI, NFI, NNFI and IFI values were greater than 0.9; $\chi^2$/df value is less than 5; SRMR and RMSEA values are in line with scholars suggest the standard. The values are: $\chi^2$/df = 3.36, RMSEA = 0.078, SRMR = 0.064, GFI = 0.90, CFI = 0.91, NFI = 0.90, NNFI = 0.94, and IFI = 0.91. Therefore, this study patterns of fit measure up to standard. See Fig 2.

**4.2.2. Structural equation modeling.** Then we use AMOS to map out this study's significant structural equation model. See Fig 2. The correlation coefficient between the usual

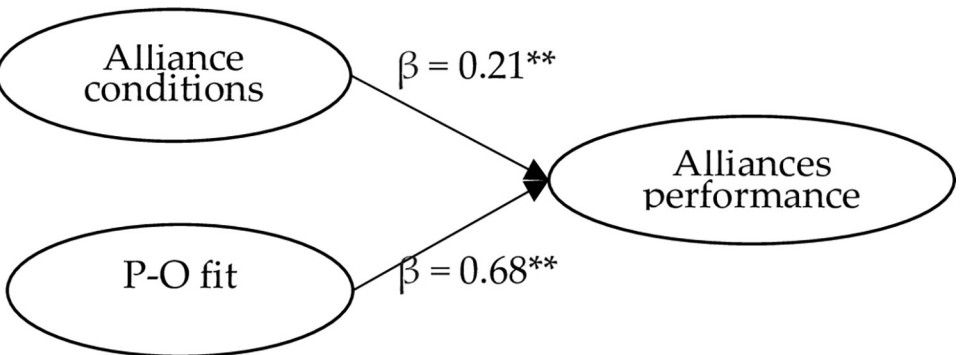

**Fig 3. The main structural equation model diagram.** $^{**}$ $p < .01$.

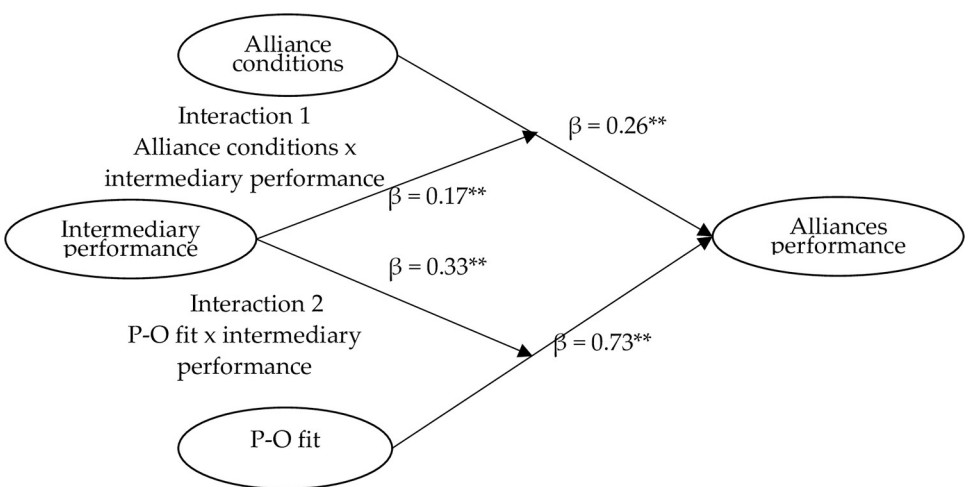

**Fig 4. The moderated structural equation model diagram.** $^{**}$ $p < .01$.

indicators, intermediary performance and alliance situation, P-O fit, and alliance performance are positively correlated.

The model-appropriate indexes, CFI, GFI, AGFI, NFI, NNFI, and IFI, were greater than 0.9 through the confirmatory factor. $\chi^2$/df value of less than 5. SRMR and RMSEA values are compliant scholars suggest [59, 65]. The specific values are: $\chi^2$/df = 3.36, RMSEA = 0.078, SRMR = 0.064, GFI = 0.90, CFI = 0.95, NFI = 0.92, NNFI = 0.94, and IFI = 0.95. Therefore, the present study shows a good fit for the structural equation model. By AMOS, the coefficients of alliance conditions to alliance performance (0.21) and P-O fit to alliance performance (0.68) both illustrate the positive impacts. Therefore, H1 and H2 are supported. See Fig 3.

### 4.3. Moderating effect of intermediary performance

Estimating the moderated model in Fig 4, the results are indicating an acceptable fit. The addition of moderator intermediary performance provided greater predictive power. The parameter values with their significance presented in the moderated model in Fig 6 revealed that intermediary performance significantly moderated the causal relations of alliance conditions→alliances performance (0.26) and P-O fit→alliance performance (0.73). Therefore, H3 and H4 were supported.

To further cement the moderation results, this study uses hierarchical regression analysis to test the moderating effect of the intermediary performance. Each variable item is standardized. Standardized variables X is the independent variable, and the manipulated variable M both are converted into Z scores, a mean centering technology, to avoid collinearity [66, 67]. Specific values as shown in Table 3.

Table 3 and Fig 5 show the influence of intermediary performance on the relationship between alliance conditions, and alliance performance has a moderating role. Therefore, H3 was supported. Also, as seen in Table 4 and Fig 6, the influence of intermediary performance on the relationship between person-organization fit and alliance performance has a moderating role. Therefore, H4 was supported. These results suggest that the relationship between alliance conditions and alliances performance and the relationship between P-O fit and alliances performance are moderated by intermediary performance.

**Table 3. Regression coefficients of alliance conditions and intermediary performance (The dependent variable: alliances performance).**

| Standardized variables | Regression coefficients | Sig. | Constant |
|---|---|---|---|
| Z alliance conditions | 0.493 | 0.000 | 3.599 |
| Z intermediary performance | 0.250 | 0.471 | |
| Z alliance conditions ×Z intermediary performance | 0.158 | 0.000 | |

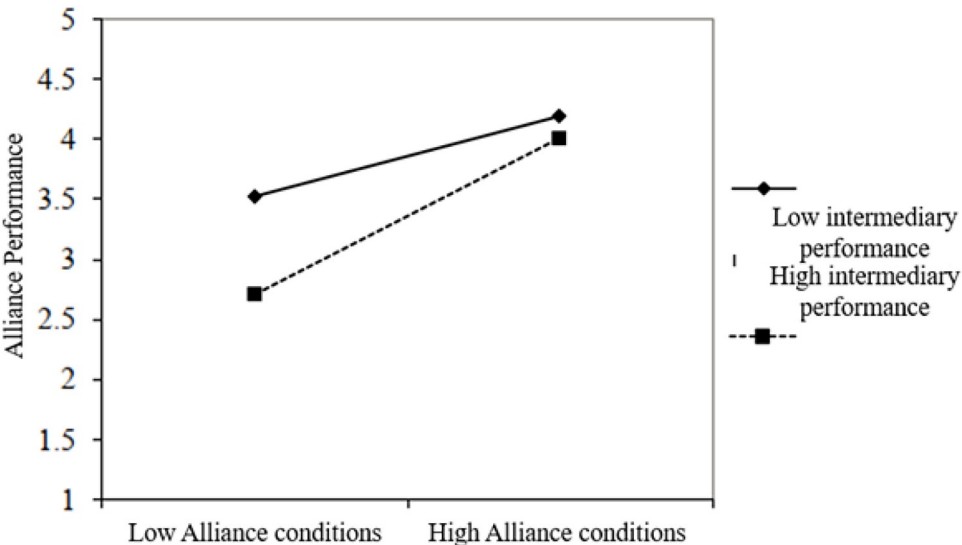

**Fig 5. The relationship of intermediary performance to alliance conditions and alliance performance.**

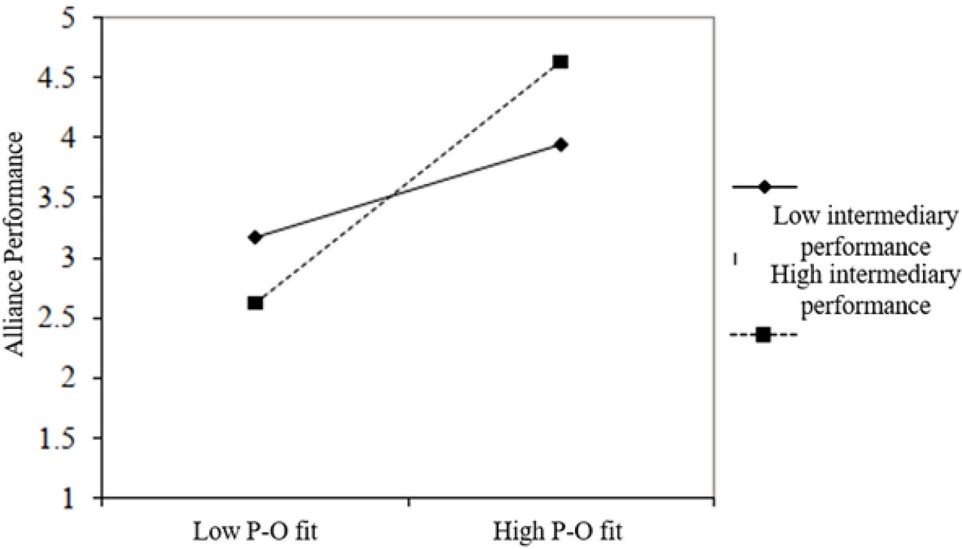

**Fig 6. The relationship of intermediary performance to person-organization fit and alliance performance.**

**Table 4. Regression coefficients of P-O fit and intermediary performance (The dependent variable: alliances performance).**

| Standardized variables | Regression coefficients | Sig. | Constant |
|---|---|---|---|
| Z p-o fit | 0.696 | 0.000 | 3.592 |
| Z intermediary performance | 0.036 | 0.196 | |
| Z p-o fit ×Z intermediary performance | 0.310 | 0.257 | |

## 5. Discussion

To sum up the statistics of this research, Table 5 provides the hypotheses results. H1, H2, H3, and H4 all obtained significant statistical support.

Many organizational behavior theories believe that a good fit between individuals and organizations is essential [14, 44, 47]. This study established a research framework through literature review, collected 406 valid questionnaires, and statistical results supported all empirical analysis hypotheses.

After empirical analysis, alliance performance-related conditions significantly affect the outcomes. It could be believed that when the characteristics of the alliance conditions are more stable, it will impact the cooperation with the alliance partners and affect the alliance's performance. This result is in line with previous studies such as [26]. So, the strategic alliance should enhance the partners' cooperation status to get more alliance performance. The higher the personal-organization fit of the organization, the higher the performance of alliance cooperation. Although there is no previous discussion of this impact path, this result still reflects its importance.

This study shows that the intermediary performance significantly correlates with the alliance conditions. And the data of intermediary performance, alliance conditions, and alliance performance can be seen after the EXCEL mapping of the intermediary performance moderating effect. After empirical analysis, it is shown that the intermediary performance has a positive moderating effect on the relationship between alliance conditions and performance. This is also a new empirical result in the academic field. We could suggest that the firms pay attention to this dimension.

## 6. Conclusions

Based on the above research conclusions, this research puts forward the following operational and management implications, which are hoped to provide industry players with some strategic directions for reference, as follows.

### 6.1 Theoretical implication

This study explored the state of the inter-league travel industry, individual-organization fit and performance of the agency relationship, and empirical research to explore the impact on

**Table 5. Summary of the hypotheses results.**

| Hypothesis | Content | Result |
|---|---|---|
| H1 | The characteristics of alliance conditions positively affect alliance performance. | Supported. |
| H2 | Person-organization fit positively affects alliance performance. | Supported. |
| H3 | The high intermediary performance will moderate the positive relationship between alliance conditions and alliance performance. | Supported. |
| H4 | The high intermediary performance will moderate the positive relationship between person-organization fit and alliance performance. | Supported. |

the alliance performance. Summed up the following conclusions based on the theoretical implications:

First, this research also uses this theory, but the conditions that occur in the alliance should not be limited to these three characteristics. Therefore, follow-up studies can supplement the characteristics of the alliance by conducting other studies on the status of the alliance.

Second, the person-organization fit has a positive impact on alliance performance. In the past, researchers have done sufficient research on this relationship. Based on the relationship between the two are directly positive, then it is also possible to conduct research on whether alliance performance positively affects person-organization fit. Further, in the past, most studies only considered the direct impact of intermediary performance on work performance, but never considered the research on the effect of intermediary performance as a moderating variable. In this research, we verified the moderating role of intermediary performance. In addition, the measurements of intermediary performance in this study are adopted from foreign scholars. Although the reliability is good, its applicability in China remains to be developed by subsequent researchers.

## 6.2 Management implications

First, pay attention to the practice and influence of the alliance conditions. When senior managers in the travel industry make decisions about strategic alliance partners, they consider the alliance conditions as a factor. Most of them only consider the characteristics of the alliance partners. They do not consider the elements of the alliance conditions that will occur in the future (aggregation intensity, conflicts between partners, interdependence). Because when alliances have sufficient resources and operating capabilities, they will get better and cheaper upstream supply, higher productivity, and more competitive product prices. Therefore, it is recommended that alliances share resources to achieve better alliance performance. However, both parties in the alliance will inevitably have tensions, incompatibility, and conflicts in their work. If they are left unchecked, the alliance will likely be unable to proceed smoothly. In addition, not having enough dependence on alliance partners can also lead to the collapse of trust between the two parties, leading to the alliance's failure. Therefore, it is recommended that alliance decision-makers fully consider the alliance situation with alliance partners when choosing alliance partners and solve the situation crisis promptly to promote the continuation of the alliance and improve business performance.

Second, a partnership with the partner's organization fit. When forming a cooperative relationship with alliance partners, it is necessary to consider whether their organizational culture, goals and inter-organizational supply and demand viewpoints are consistent with their organizations. Because real teamwork, respect for trust, and the orientation and management of work goals can reduce conflicts, reduce the time cost of mutual running-in, and make the alliance work more smoothly. And give full play to the professional knowledge and experience required by the alliance work, meet each other's needs, have a good working environment, improve work efficiency, obtain good organizational performance, and promote the development and continuation of the alliance. Therefore, besides measuring the professional knowledge and capabilities of the alliance partners and other hardware facilities, it is also necessary to consider whether their values, goals, needs and other software are consistent with their organizations as the selection criteria when selecting alliance partners. The education and training organized by the alliance improve the fit of the employees' values in the alliance.

Third, in the vein of to improve the marketing management of agency performance, we suggest follows. Close overseas contacts, rich industry experience, better financial capabilities, higher negotiation skills, and experienced supervisors will enable the organization to gain a

leading role in the same industry and products, enabling the organization to have a strong backing. Therefore, it is recommended that organizations form excellent marketing capabilities and sales strategies. The sales manager should lobby the management to reserve a budget for training the sales team. The use of marketing skills is a manifestation of marketing capabilities. In addition, managers need to organize the sales team's market research skills to understand market trends and new products and services and improve the performance of intermediaries. Sales managers need to ensure effective pricing, the scope of sales of new products, channel relationship management, and promotion training because these form-critical aspects of the sales team's marketing capabilities. Secondly, sales managers should lobby the administration to approve performance compensation plans, bonuses, and incentive plans. These will motivate sales staff to practice daily goals to ensure that customer and retailer information is effectively used, which will promote higher intermediary performance. Such a move can see the profit contribution area, and the market has better reassessment and capture. Under normal circumstances, applying these two marketing capabilities at the same time and appropriately assigning strategies will achieve the goal of enhancing organizational performance.

### 6.3 Limitations and suggestions

The findings are based on research-oriented objectives and the architecture of the interior of China travel industry professionals as the research object. The main content is to study league status, person-organization fit, and the relationship between the agency and the alliance performance. This research cannot fully cover rigorous and objective methods, limited to the following factors.

This research adopts the method of purposeful sampling. It takes travel industry practitioners as the research sample, and the selected travel industry-related industries are many. Still, no specific analysis has been made, and there may be problems that have not yet been discussed in depth. Although it refers to scholars' literature and is based on existing scales, the questionnaire is measured in a self-evaluation method. It is impossible to confirm whether the subject has grasped the original intent of the questionnaire and expresses a valid opinion, which may cause an error in the results. Since strategic alliances are a two-sided matter, performance measurement must also consider the perception of alliance partners. However, due to human resources and time constraints, it is impossible to understand the opinions and ideas of alliance partners, so it may also affect the analysis results.

We suggest the following comments:

First, this study adopts the questionnaire survey method, which may produce the respondent's defensive and reserved attitude, which will affect the results. There are still shortcomings, aiming at the homology prevention measures of the interviewed information concealment method, the item concealment method, and the topic random allocation method. Therefore, it is recommended that subsequent researchers design questionnaires using matching questionnaires. More complete data can improve the questionnaire's objectivity and reduce the homology error.

Second, the research object is limited to travel industry practitioners in mainland China to conduct empirical analysis. It is suggested that follow-up researchers include travel industry practitioners from other countries or regions in the research to explore whether there are differences.

Third, evaluate and discuss alliance performance with more theoretical perspectives. Many scholars have different views on the discussion indicators of alliance performance in their research. This study only examines the performance of travel industry alliances in terms of the

upstream supply situation, productivity, and product prices. It is suggested that follow-up research can find different aspects of the performance evaluation of travel industry alliances. Discuss to provide industry players with suggestions on choosing partners and improving alliance performance.

## Supporting information

**S1 Data.**
(SAV)

## Author Contributions

**Conceptualization:** Hui Sun, You-Yu Dai.

**Data curation:** Chaochen Zhang, Rok Lee.

**Formal analysis:** Hui Sun, You-Yu Dai.

**Investigation:** Hui Sun, You-Yu Dai, Chaochen Zhang.

**Methodology:** Hui Sun, You-Yu Dai.

**Project administration:** Jin-Hua Chu.

**Resources:** You-Yu Dai.

**Software:** Hui Sun, You-Yu Dai.

**Supervision:** Jin-Hua Chu.

**Validation:** Hui Sun, You-Yu Dai.

**Visualization:** You-Yu Dai.

**Writing – original draft:** Hui Sun, You-Yu Dai.

**Writing – review & editing:** Su-Sung Jeon.

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
