## [Decision Letter · Decision Letter 0]

24 May 2022

PONE-D-22-08099Moderating Effect of Intermediary Performance on the Performance in Travel Industry Between Strategic Alliance and Person-Organization FitPLOS ONE

Dear Dr. Dai,

Thank you for submitting your manuscript to PLOS ONE. After careful consideration, we feel that it has merit but does not fully meet PLOS ONE’s publication criteria as it currently stands. Therefore, we invite you to submit a revised version of the manuscript that addresses the points raised during the review process.

We look forward to receiving your revised manuscript.

Kind regards,

María del Carmen Valls Martínez, Ph.D.

Academic Editor

PLOS ONE

Journal Requirements:

3. Thank you for stating the following in the funding Section of your manuscript:

“: Special project for construction theory of applied undergraduate university in Shandong Jiaotong University, grant number YYX20211204; The National Social Science Fund of China, grant number 20BGL219.”

We note that you have provided additional information within the funding Sectio. Please note that funding information should not appear in the Acknowledgments section or other areas of your manuscript. We will only publish funding information present in the Funding Statement section of the online submission form.

“You-Yu Dai

Grant number YYX20211204

Special project for construction theory of applied undergraduate university in Shandong Jiaotong University

http://fzghc.sdjtu.edu.cn/

Jinhua Chu

Grant number 20BGL219

The National Social Science Fund of China

http://www.nopss.gov.cn/

Reviewers' comments:

Reviewer's Responses to Questions

**Comments to the Author**

1. Is the manuscript technically sound, and do the data support the conclusions?

Reviewer #1: Yes

Reviewer #2: Partly

Reviewer #3: Yes

2. Has the statistical analysis been performed appropriately and rigorously? 

Reviewer #1: Yes

Reviewer #2: Yes

Reviewer #3: Yes

3. Have the authors made all data underlying the findings in their manuscript fully available?

Reviewer #1: Yes

Reviewer #2: Yes

Reviewer #3: Yes

4. Is the manuscript presented in an intelligible fashion and written in standard English?

Reviewer #1: Yes

Reviewer #2: No

Reviewer #3: Yes

5. Review Comments to the Author

Reviewer #1: Dear Authors

Thank you for giving me the opportunity to read your article. I have decided a minor revision. However, you should pay attention to the comments I indicate.

1. Introduction

It is necessary to clearly indicate the objective, the need for the research and especially the contribution in the introduction section.

2. Discussion of results

It is necessary to enlarge and redo section. The results should be confirmed or not using the literature.

3.I propose to change the title of the section "Implications and Further Suggestion" to "Conclusions". In addition, this section should be reordered, with the Theoretical Implications in the first place "5.1".

I hope the commments were usefull.

Best regards

Reviewer #2: I appreciate you for the selection of this topic. This is novel topic. However, the authors need to incorporate some suggestions that can improve the quality of paper. Specific comments are provided separately.

Reviewer #3: The paper “Moderating Effect of Intermediary Performance on the Performance in Travel Industry Between Strategic Alliance and Person-Organization Fit” analyses the effects of alliance conditions and p-o-fit on the performance of strategic alliances between travel industries using a structural equation modeling method with 406 observations. The paper is well structured, has a good review of literature and has a good empirical development, however, there are several minor a major questions that the authors must solve in order to improve the paper:

Major questions:

1. In the section 2.3, the authors must show clearer the motivation to consider the moderating effects because it is no clear. Moreover, they must show if there are others paper that considers these effects in other sectors different of travel industry.

2. In the section 3.2., the authors must show clearer the kind of personnel in travel industries that they answer the questionary. Moreover, in this section is necessary an indication that the items used are show in table 1. Are there not more items to measure the alliance performance? Maybe 3 are few.

3. In the section 3.3, it is necessary a frequency table that show the different categories and the percentage of answers. Moreover, a table with KMO and MSA values is necessary.

4. The most important questions are about section 4.3. Why the authors use a structural equation model and to analyses the moderating effect a hierarchical regression analysis? This have not sense and maybe is more interesting analyses the moderating effects using a structural equation model or PLS.

5. Moreover, these results are the most important taking into account the paper title and that in the abstract the authors emphasize these moderating effects, however, the results are presents briefly and without show the most important aspect of the regression developed.

Minor questions;

1.- In the abstract appears p-o fit but the significance of this abbreviation appears in the page.

2. Maybe be in the title appears too many times performance.

3. The authors must include several references about papers that analyses the moderating effects in the travel industries or others.

6. PLOS authors have the option to publish the peer review history of their article (what does this mean?). If published, this will include your full peer review and any attached files.

Reviewer #1: No

Reviewer #2: No

Reviewer #3: No

---

## [Author Response · Author response to Decision Letter 0]

13 Sep 2022

Response Letter

Journal Title: Plos One

Manuscript ID: PONE-D-22-08099

Manuscript Title: Moderating Effect of Intermediary Performance on the Performance in Travel Industry Between Strategic Alliance and Person-Organization Fit

We sincerely appreciate all the detailed comments and suggestions made by the reviewers. All comments are considered carefully and summarized as follows, and all the revisions are marked in Red in the original paper for reviewers’ readability. In addition, the revised manuscript has been sent to Editage Editing Services for cosmetic editing. We believe the presentation of this revised manuscript should be more expressive and more readable than the original manuscript. Please do not hesitate to give us any comments and we also look forward to your further decision.

Responses from the authors to Reviewer 1:

Reviewer #1: 

Dear Authors

Thank you for giving me the opportunity to read your article. I have decided a minor revision. However, you should pay attention to the comments I indicate.

Response: Thank you for your professional and attentive comments, we revised it to follow your comments. 

1. Introduction

It is necessary to clearly indicate the objective, the need for the research and especially the contribution in the introduction section.

Response: Thanks for your rich comments. We added the need as the following:

Therefore, more and more enterprises participate in strategic alliances, hoping to increase the scope of operations and gain a comparative advantage in the industry, thereby reducing costs, increasing operating performance or firm performance, resource and knowledge sharing, and improving profitability (Stouthuysen et al., 2017; Lee, 2019; Cacciolatti et al., 2020; Emami et al., 2022).

We added the contribution as the following:

Meanwhile, In the past, most studies only considered the direct impact of intermediary performance on work performance. Still, they never felt the research on intermediary performance's effect as a moderating variable. In this research, we verified the moderating role of intermediary performance. Further, we consider the P-O fit factor to affect alliance performance.

References

1.Lee ES. Improving a firm's performance within a strategic shipping alliance. The Asian Journal of Shipping and Logistics. 2019; 35(4): 213-219.

2.Cacciolatti L, Rosli A, Ruiz-Alba JL, Chang J. Strategic alliances and firm performance in startups with a social mission. Journal of Business Research. 2020; 106: 106-117.

3.Emami A, Welsh DH, Davari A, Rezazadeh A. Examining the relationship between strategic alliances and the performance of small entrepreneurial firms in telecommunications. International Entrepreneurship and Management Journal. 2022; 18(2): 637-662.

2.Discussion of results

It is necessary to enlarge and redo the section. The results should be confirmed or not using the literature.

Response: We agree with your opinion. In the discussion part, we deleted some references and modified them. The modified results are as follows.

4.4Discussion

To sum up the statistics of this research, Table 5 provides the hypotheses results. H1, H2, H3, and H4 all obtained significant statistical support.

Table 5. Summary of the hypotheses results

Hypothesis Content Result

H1 The characteristics of alliance conditions positively affect alliance performance. Supported.

H2 Person-organization fit positively affects alliance performance. Supported.

H3 The high intermediary performance will moderate the positive relationship between alliance conditions and alliance performance. Supported.

H4 The high intermediary performance will moderate the positive relationship between person-organization fit and alliance performance. Supported.

Many organizational behavior theories believe that a good fit between individuals and organizations is essential (e.g., Chen et al., 2020; Hamidah et al., 2017; Siyal et al., 2020). This study established a research framework through literature review, collected 406 valid questionnaires, and statistical results supported all empirical analysis hypotheses. 

After empirical analysis, alliance performance-related conditions significantly affect the outcomes. It could be believed that when the characteristics of the alliance conditions are more stable, it will impact the cooperation with the alliance partners and affect the alliance's performance. This result is in line with previous studies such as Das and Teng (2002). So, the strategic alliance should enhance the partners' cooperation status to get more alliance performance. The higher the personal-organization fit of the organization, the higher the performance of alliance cooperation. Although there is no previous discussion of this impact path, this result still reflects its importance.

This study shows that the intermediary performance significantly correlates with the alliance conditions. And the data of intermediary performance, alliance conditions, and alliance performance can be seen after the EXCEL mapping of the intermediary performance moderating effect. After empirical analysis, it is shown that the intermediary performance has a positive moderating effect on the relationship between alliance conditions and performance. This is also a new empirical result in the academic field. We could suggest that the firms pay attention to this dimension.

References

1.Chen X, Liu M, Liu C, Ruan F, Yuan Y, Xiong C. Job satisfaction and hospital performance rated by physicians in China: A moderated mediation analysis on the role of income and person-organization fit. International Journal of Environmental Research and Public Health. 2020; 17(16): 5846.

2.Hamidah H, Mukhtar M, Karniati N. The effect of person-organization fit, job satisfaction, and trust toward high schools’ teachers affective commitment. Indonesian Journal of Educational Review. 2017; 4(2): 26-38.

3.Siyal S, Xin C, Peng X, Siyal AW, Ahmed W. Why do high-performance human resource practices matter for employee outcomes in public sector universities? The mediating role of person-organization fit mechanism. SAGE Open. 2020; 10(3): 2158244020947424.

3. I propose to change the title of the section "Implications and Further Suggestion" to "Conclusions". In addition, this section should be reordered, with the Theoretical Implications in the first place "5.1".

Response: We changed the title of the section "Implications and Further Suggestion" to "Conclusions". And the title of the corresponding collation and summary

5. Conclusions

5.1 Theoretical implication

5.2. Management implications

I hope the comments were useful.

Best regards

Response: Many thanks again to your suggestions.

Responses from the authors to Reviewer 2:

Reviewer #2: 

I appreciate you for the selection of this topic. This is a novel topic. 

Response: Thank you for your praise and precious comments.

However, the authors need to incorporate some suggestions that can improve the quality of the paper. Specific comments are provided separately.

Response: Thank you for your praise and precious comments. We have revised and improved it according to your opinions. We have revised and improved it according to your comments. 

1. The introduction section can be improved with standard English writing.

Response: Thank you for your comment. The revised manuscript has been sent to Editage Editing Services for cosmetic editing.

2. Second paragraph first line introduction citation is missing. Kindly cite the first sentence. 

Response: Thank you for your comment. We have added the citation.

However, alliances are often accompanied by a higher rate of alliance failures, and researches on the problems of alliance performance have always been a difficult and hot point in alliances-related studies (Bleeke, & Ernst, 1991; Fischer et al., 2021).

References

1.Bleeke J, Ernst D. The way to win in cross-border alliances. Harvard Business Review. 1991; 69(6): 127-135.

2.Fischer D, Greven A, Tornow M, Brettel M. On the value of effectuation processes for R&D alliances and the moderating role of R&D alliance experience. Journal of Business Research. 2021; 135: 606-619.

3. Fourth paragraph first sentence citation is missing. Kindly cite this sentence. 

Response: Thank you for your comment. We have added the citation.

Nowadays, the various industries such as airlines, travel services, telecommunications, software, hardware and educational services and automotive industries required alliance as key growth strategies to improve performance (Hsu & Prescott, 2016; Reuer & Arino, 2007).

References

1.Hsu ST, Prescott JE. The alliance experience transfer effect: The case of industry convergence in the telecommunications equipment industry. British Journal of Management. 2016; 28(3): 425–443. 

2.Reuer J, Arino A. Strategic alliance contracts: Dimensions and determinants of contractual complexity. Strategic Management Journal. 2007; 28(3): 313–330. 

4. Clearly explain the gap in the current study in the introduction section second last paragraph and explain how this study contributes to the existing literature.

Response: Many thanks for your valuable comment. We added the following paragraph to explain the research gap.

Meanwhile, in the past, most studies only considered the direct impact of intermediary performance on work performance, but never considered the research on the effect of intermediary performance as a moderating variable. In this research, we verified the moderating role of intermediary performance. Further, we consider P-O fit factor to affect alliance performance. 

5. Literature review 

5.1 section 2.1. First paragraph citations are missing. Kindly cite the relevant studies.

Response: Thank you for your comment. We have added the citation.

Accordingly, the positive factors of the collective strength, overall resources, or competitiveness of the enterprises in the alliance contribute to the alliance's performance (Huda et al., 2019). In other words, the alliance strategy can achieve goals through the power of joint partners. The resources of a single enterprise will not always be beneficial with the help of the strategic alliance's collective partners. It can also ensure that the alliance can achieve performance. In related research, the collective strength of the partnership is positively correlated with alliance performance (Taylor et al., 2018; He et al., 2020).

References

1.He Q, Meadows M, Angwin D, Gomes E, Child J. Strategic alliance research in the era of digital transformation: Perspectives on future research. British Journal of Management. 2020; 31(3): 589-617.

2.Huda M, Qodriah SL, Rismayadi B, Hananto A, Kardiyati EN, Ruskam A, Nasir BM. Towards cooperative with competitive alliance: Insights into performance value in social entrepreneurship. In Creating business value and competitive advantage with social entrepreneurship. IGI Global. 2019. p. 294-317.

3.Taylor EC, McLarty BD, Henderson DA. The fire under the gridiron: resource dependence and NCAA conference realignment. Journal of Business Research. 2018; 82: 246-259.

5.2 Section 2.2. Again discussion without citations. Kindly add relevant citations.

Response: Thank you for your comment. We have added some citations to the discussion.

Many organizational behavior theories believe that a good fit between individuals and organizations is essential (e.g., Chen et al., 2020; Hamidah et al., 2017; Siyal et al., 2020). This study established a research framework through literature review, collected 406 valid questionnaires, and statistical results supported all empirical analysis hypotheses. 

After empirical analysis, alliance performance-related conditions significantly affect the outcomes. It could be believed that when the characteristics of the alliance conditions are more stable, it will impact the cooperation with the alliance partners and affect the alliance's performance. This result is in line with previous studies such as Das and Teng (2002). So, the strategic alliance should enhance the partners' cooperation status to get more alliance performance. The higher the personal-organization fit of the organization, the higher the performance of alliance cooperation. Although there is no previous discussion of this impact path, this result still reflects its importance.

This study shows that the intermediary performance significantly correlates with the alliance conditions. And the data of intermediary performance, alliance conditions, and alliance performance can be seen after the EXCEL mapping of the intermediary performance moderating effect. After empirical analysis, it is shown that the intermediary performance has a positive moderating effect on the relationship between alliance conditions and performance. This is also a new empirical result in the academic field. We could suggest that the firms pay attention to this dimension.

References

1. Chen X, Liu M, Liu C, Ruan F, Yuan Y, Xiong C. Job satisfaction and hospital performance rated by physicians in China: A moderated mediation analysis on the role of income and person-organization fit. International Journal of Environmental Research and Public Health. 2020; 17(16); 5846.

2. Hamidah H, Mukhtar M, Karniati N. The effect of person-organization fit, job satisfaction, and trust toward high schools’ teachers affective commitment. Indonesian Journal of Educational Review. 2017; 4(2): 26-38.

3. Siyal S, Xin C, Peng X, Siyal AW, Ahmed W. Why do high-performance human resource practices matter for employee outcomes in public sector universities? The mediating role of person-organization fit mechanism. SAGE Open. 2020; 10(3): 2158244020947424.

5.3 Section 2.3. authors discuss moderating effect of intermediary performance with only 3 citations in three paragraph. Kindly cite the justifications provided in the second and third paragraphs.

 Response: Thank you for your comment. We have added some citations to the section 2.3.

2.3 The moderating influence of intermediary performance

Travel enterprises must deal with export and import business, like trade enterprises. So, it is known as inbound and outbound travel. Quartey and Oguntoye (2020) indicated that studies narrowly focus on intermediary performance and its antecedents and consequences. Improving intermediary performance is the key determinant in promoting corporate sustainability. Despite the extensive scholarly research on strategic alliances (Shah & Swaminathan, 2008), previous research has not considered intermediary performance as a moderator of the impact of various factors influencing alliance performance. This study innovatively adopts intermediary performance from the trade field to examine its role in the travel industry. There is no specific unified definition of intermediary performance. Referring to Suwannarat (2016), intermediary performance in this study is the performance of a travel-intermediary-related firm. The previous researchers only test the direct causal relationship of intermediary performance. Suwannarat (2016) measured intermediary performance by acquiring new clients, retaining existing clients, goal achievement, market share, income, and export growth. Khasanah and Wicaksono (2021) empirically tested the causal relationship between economic growth and intermediary performance. 

Peng and York (2001) divided the intermediary performance from the perspective of intermediaries into search costs, negotiation costs, and monitoring or implementation costs. The search cost generally involves acquiring knowledge through market research and planning in advance, and obtaining such knowledge without external help can be expensive and time-consuming (Ibeh et al., 2019). High search costs not only prevent many importers and exporters of the tourism industry from expanding internationally, but they may also lead to vague searches in the past, thereby increasing the possibility of bad debts in the cost and affecting exports (Huang et al., 2020). At this time, travel agencies can help, by providing experience in understanding foreign markets and export processes, so that the alliance's tourism industry players can be familiar with international marketing strategies. 

Negotiation costs include the direct costs of travel and personnel and the ex-ante costs of dealing with potential dangers with unfamiliar foreign customers (Qi et al., 2020). Tourism imports and export companies may lack experience in foreign markets in this negotiation. Because of their lack of cultural understanding, the complexity of negotiation norms is derived. Because of their expertise, tourism agencies will assist the alliance's tourism industry players in reducing negotiation costs.

References

1.Huang H, Liu Y, Lu D. Proposing a model for evaluating market efficiency of OTAs: Theoretical approach. Tourism Economics. 2020; 26(6): 958-975.

2.Ibeh K, Crick D, Etemad H. International marketing knowledge and international entrepreneurship in the contemporary multi speed global economy. International Marketing Review. 2019.

3.Qi X, Chan JH, Hu J, Li Y. Motivations for selecting cross-border e-commerce as a foreign market entry mode. Industrial Marketing Management. 2020; 89: 50-60.

4.Quartey SH, Oguntoye O. Promoting corporate sustainability in small and medium‐sized enterprises: Key determinants of intermediary performance in Africa. Business Strategy and the Environment. 2020; 29(3): 1160-1172.

5.Shah RH, Swaminathan V. Factors influencing partner selection in strategic alliances: The moderating role of alliance context. Strategic Management Journal. 2008; 29(5): 471-494.

Chapter 3.

3.1 Research model: it is better to discuss about the relationships among construct in literature review chapter and insert the figure at the end of 2nd chapter. 

Response: Thank you for your comment, and we totally agree with you. We have moved the original section 3.1 to 2.4 as follows.

2.4 Research Model

3.2. Questionnaire: What do you mean by travel related industries? Kindly explain the related industries.

Response: Thank you for your comment. We added the explanation as follows. And the details are shown in Table 1.

The travel-related industries include such as travel agencies, airline companies, hotels, and hospitality stores.

Table 1. Demographic statistics

Personal attributes Classification Number of samples Proportion (%) Personal attributes Classification Number of samples Proportion (%)

gender Male 193 47.5 Education level High school 30 7.4

 Female 213 52.5 College 63 15.5

age 18-25 83 20.4 University 253 62.3

 26-30 123 30.3 graduate 60 14.8

 31-40 137 33.7 position intern 40 9.9

 41-50 39 9.6 Ordinary employees 122 30.0

 51 and above 24 5.9 Grassroots managers 104 25.6

Marriage status Married 248 61.1 Middle managers 117 28.8

 Unmarried 157 38.7 Top managers 23 5.7

Work years 1-3 120 29.6 Travel-related industry Catering 66 16.3

 4-6 112 27.6 Accommodation 17 4.2

 7-9 72 17.7 Transportation 68 16.7

 10 and above 102 25.1 Travel industry (such as travel agency) 37 9.1

Monthly income (RMB) Less than 6000 159 39.2 Cultural and entertainment industry (such as scenic spots, museum, park) 78 19.2

 6001-8000 119 29.3 Tourism and shopping 27 6.7

 8001-10000 83 20.4 Other travel-related industries 113 27.8

 More than 10000 45 11.0 

3.3. Data: What was the original sample size? How you determine the sample size? How many questionnaires were distributed. Kindly explain above points. 

Response: Thank you for your comment. We added the explain as follow.

This study was conducted from December 30 to January 15, 2021, and formal questionnaires were distributed through the incentive online questionnaire survey. A total of 480 questionnaires were distributed, and 455 questionnaires were recovered, with a recovery rate of 92%. The recovered samples were screened, and the invalid questionnaires (such as incomplete questionnaires) were deleted. A total of 406 questionnaires were effectively recovered, with an effective recovery rate of 89%. The number of valid questionnaires meets the criteria of more than 10 times the number of questionnaire items (Dai et al., 2020).

References

1.Dai YY, Shie AJ, Liu ZJ. How service industry attract employee? Evidence from website quality. International Journal of Industrial and Systems Engineering. 2020; 36(1): 125-148.

4. Results

4.1 Since the study data collected was self-reported. Did you check common method bias? If yes, then how?

Response: Thanks for your valuable question. We have checked common method bias, as follows.

In this study, anonymous responses were set to control for possible common method bias, and the common method bias was tested using the Harman single factor method (Qian et al., 2020). The results indicated that no serious common method bias existed in this study.

References

1. Qian Y, Chen F, Yuan C. The effect of co-parenting on children’s emotion regulation under fathers’ perception: A moderated mediation model of family functioning and marital satisfaction. Children and Youth Services Review. 2020; 119: 105501.

4.2 Table 1:

Alliance conditions standard factor laoding values are low when compared intermediary performance SFL but its CR value is 0.92. Contrary to this the CR value of intermediary performance is low with high standard factor loadings. How can you justify?

Response: Thank you very much for correcting. We have recalculated the CR and AVE values. Please see Table 2.

4.3 Kindly add table 4 summarise the hypotheses results. And also discuss individual hypothesis result and compare with past studies for better understanding of the readers. Extend and improve discussion section. In the current form discussion section is very weak. 

Response: We are very thankful for your rich comments. We have extended the discussion section.

4.4 Discussion

To sum up the statistics of this research, Table 5 provides the hypotheses results. H1, H2, H3, and H4 all obtained significant statistical support.

Table 5. Summary of the hypotheses results

Hypothesis Content Result

H1 The characteristics of alliance conditions positively affect alliance performance. Supported.

H2 Person-organization fit positively affects alliance performance. Supported.

H3 The high intermediary performance will moderate the positive relationship between alliance conditions and alliance performance. Supported.

H4 The high intermediary performance will moderate the positive relationship between person-organization fit and alliance performance. Supported.

Many organizational behavior theories believe that a good fit between individuals and organizations is essential (e.g., Chen et al., 2020; Hamidah et al., 2017; Siyal et al., 2020). This study established a research framework through literature review, collected 406 valid questionnaires, and statistical results supported all empirical analysis hypotheses. 

After empirical analysis, alliance performance-related conditions significantly affect the outcomes. It could be believed that when the characteristics of the alliance conditions are more stable, it will impact the cooperation with the alliance partners and affect the alliance's performance. This result is in line with previous studies such as Das and Teng (2002). So, the strategic alliance should enhance the partners' cooperation status to get more alliance performance. The higher the personal-organization fit of the organization, the higher the performance of alliance cooperation. Although there is no previous discussion of this impact path, this result still reflects its importance.

This study shows that the intermediary performance significantly correlates with the alliance conditions. And the data of intermediary performance, alliance conditions, and alliance performance can be seen after the EXCEL mapping of the intermediary performance moderating effect. After empirical analysis, it is shown that the intermediary performance has a positive moderating effect on the relationship between alliance conditions and performance. This is also a new empirical result in the academic field. We could suggest that the firms pay attention to this dimension.

References

1.Chen X, Liu M, Liu C, Ruan F, Yuan Y, Xiong C. Job satisfaction and hospital performance rated by physicians in China: A moderated mediation analysis on the role of income and person-organization fit. International Journal of Environmental Research and Public Health. 2020; 17(16): 5846.

2.Hamidah H, Mukhtar M, Karniati N. The effect of person-organization fit, job satisfaction, and trust toward high schools’ teachers affective commitment. Indonesian Journal of Educational Review. 2017; 4(2): 26-38.

3.Siyal S, Xin C, Peng X, Siyal AW, Ahmed W. Why do high-performance human resource practices matter for employee outcomes in public sector universities? The mediating role of person-organization fit mechanism. SAGE Open. 2020; 10(3): 2158244020947424.

Responses from the authors to Reviewer 3:

Reviewer #3: 

The paper “Moderating Effect of Intermediary Performance on the Performance in Travel Industry Between Strategic Alliance and Person-Organization Fit” analyses the effects of alliance conditions and p-o-fit on the performance of strategic alliances between travel industries using a structural equation modeling method with 406 observations. 

1. The paper is well structured, has a good review of literature and has a good empirical development, however, there are several minor a major questions that the authors must solve in order to improve the paper:

Response: Thanks for your praise and good comments to improve the quality of the article.

2. Major questions:

2.1 In the section 2.3, the authors must show clearer the motivation to consider the moderating effects because it is no clear. Moreover, they must show if there are others paper that considers these effects in other sectors different of travel industry.

Response: Thanks for your valuable comment. We have added the explanation of the consideration of moderating effect. Please see as follows,

Travel enterprises must deal with export and import business, like trade enterprises. So, it is known as inbound and outbound travel. Quartey and Oguntoye (2020) indicated that studies narrowly focus on intermediary performance and its antecedents and consequences. Improving intermediary performance is the key determinant in promoting corporate sustainability. Despite the extensive scholarly research on strategic alliances (Shah & Swaminathan, 2008), previous research has not considered intermediary performance as a moderator of the impact of various factors influencing alliance performance….

References

1.Quartey SH, Oguntoye O. Promoting corporate sustainability in small and medium‐sized enterprises: Key determinants of intermediary performance in Africa. Business Strategy and the Environment. 2020; 29(3): 1160-1172.

2.Shah RH, Swaminathan V. Factors influencing partner selection in strategic alliances: The moderating role of alliance context. Strategic Management Journal. 2008; 29(5): 471-494.

2.2 In the section 3.2., the authors must show clearer the kind of personnel in travel industries that they answer the questionary. Moreover, in this section is necessary an indication that the items used are show in table 1. Are there not more items to measure the alliance performance? Maybe 3 are few.

Response: Thanks for your valuable comment. In “Table 1. Demographic statistics” we provided clear background information of the responders.

Table 1. Demographic statistics

Personal attributes Classification Number of samples Proportion (%) Personal attributes Classification Number of samples Proportion (%)

gender Male 193 47.5 Education level High school 30 7.4

 Female 213 52.5 College 63 15.5

age 18-25 83 20.4 University 253 62.3

 26-30 123 30.3 graduate 60 14.8

 31-40 137 33.7 position intern 40 9.9

 41-50 39 9.6 Ordinary employees 122 30.0

 51 and above 24 5.9 Grassroots managers 104 25.6

Marriage status Married 248 61.1 Middle managers 117 28.8

 Unmarried 157 38.7 Top managers 23 5.7

Work years 1-3 120 29.6 Travel-related industry Catering 66 16.3

 4-6 112 27.6 Accommodation 17 4.2

 7-9 72 17.7 Transportation 68 16.7

 10 and above 102 25.1 Travel industry (such as travel agency) 37 9.1

Monthly income (RMB) Less than 6000 159 39.2 Cultural and entertainment industry (such as scenic spots, museum, park) 78 19.2

 6001-8000 119 29.3 Tourism and shopping 27 6.7

 8001-10000 83 20.4 Other travel-related industries 113 27.8

 More than 10000 45 11.0 

Indeed, the three items of alliances performance is not many, however, we chosen the core items and integrated questions of alliances performance. For instances, Owing to our cooperation with each other, we get higher standards of productivity (For example, the group rate, the number of groups and the ticket number of sheets, etc.)

2.3 In the section 3.3, it is necessary a frequency table that show the different categories and the percentage of answers. Moreover, a table with KMO and MSA values is necessary.

Response: Thanks for your valuable comment. We have showed the “Table 1 Demographic statistics”.

Table 1. Demographic statistics

Personal attributes Classification Number of samples Proportion (%) Personal attributes Classification Number of samples Proportion (%)

gender Male 193 47.5 Education level High school 30 7.4

 Female 213 52.5 College 63 15.5

age 18-25 83 20.4 University 253 62.3

 26-30 123 30.3 graduate 60 14.8

 31-40 137 33.7 position intern 40 9.9

 41-50 39 9.6 Ordinary employees 122 30.0

 51 and above 24 5.9 Grassroots managers 104 25.6

Marriage status Married 248 61.1 Middle managers 117 28.8

 Unmarried 157 38.7 Top managers 23 5.7

Work years 1-3 120 29.6 Travel-related industry Catering 66 16.3

 4-6 112 27.6 Accommodation 17 4.2

 7-9 72 17.7 Transportation 68 16.7

 10 and above 102 25.1 Travel industry (such as travel agency) 37 9.1

Monthly income (RMB) Less than 6000 159 39.2 Cultural and entertainment industry (such as scenic spots, museum, park) 78 19.2

 6001-8000 119 29.3 Tourism and shopping 27 6.7

 8001-10000 83 20.4 Other travel-related industries 113 27.8

 More than 10000 45 11.0 

We also added the following paragraphs to indicate the KMO values of the constructs. Please see Section 4.

Before conducting confirmatory analysis, this study first evaluates whether the data conforms to the multivariate normality. A total of 39 questions are composed of four facets. The skewness of each question is between -0.684 and 0.106, and the absolute value is less than the standard of 3.0 (Kline, 2005; Dai et al., 2022). The kurtosis of each question is between -0.909 and 0.862, and its absolute value is less than the standard of 10.0 (Kline, 2005; Dai et al., 2022). It indicates that the questions in this study are normally distributed and can be estimated by the maximum likelihood method. Therefore, this study uses the maximum likelihood method to estimate the structural equation model.

The KMO value of the whole sample is 0.96, which indicates that it is very suitable for factor analysis (Dai et al., 2020; Dai et al., 2022). The sig value of Bartlett's sphericity test is 0.000, which is less than the significance level of 0.05. Therefore, there is a correlation between variables, which is suitable for factor analysis. According to the exploratory factor analysis results of the alliance condition scale, three factors were obtained in this study, which were named aggregation intensity, conflict, and interdependence among partners, and could explain 76.52% of the variables. The KMO is 0.85. The exploratory factor analysis results of the individual organization fit scale show that there are two dimensions, namely, complementary fit and complementary fit, which can explain 71.49% of the variables. KMO was 0.96. The exploratory factor analysis results of the intermediary performance scale have three dimensions, namely, search cost, and negotiation comonitoring / implementationntation cost, which can explain 81.77% of the variables. KMO was 0.96. The above results indicate that the alliance status, individual organization fit, and intermediary performance scale of this study have enough internal correlation in the sampling data, and can reasonably conduct exploratory factor analysis.

2.4The most important questions are about section 4.3. Why the authors use a structural equation model and to analyses the moderating effect a hierarchical regression analysis? This have not sense and maybe is more interesting analyses the moderating effects using a structural equation model or PLS. Moreover, these results are the most important taking into account the paper title and that in the abstract the authors emphasize these moderating effects, however, the results are presents briefly and without show the most important aspect of the regression developed.

Response: Thanks for your valuable question. We added the moderating effect by using SEM.

4.3. Moderating effect of intermediary performance

Estimating the moderated model in Figure 4, the results are indicating an acceptable fit. The addition of moderator intermediary performance provided greater predictive power. The parameter values with their significance presented in the moderated model in Figure 6 revealed that intermediary performance significantly moderated the causal relations of alliance conditions→alliances performance (0.26) and P-O fit→alliance performance (0.73). Therefore, H3 and H4 were supported.

Figure 4. The moderated structural equation model diagram. ** p < .01.

To further cement the moderation results, this study uses hierarchical regression analysis to test the moderating effect of the intermediary performance. Each variable item is standardized. Standardized variables X is the independent variable, and the manipulated variable M both are converted into Z scores, a mean centering technology, to avoid collinearity (Aiken & West, 1991; Kraemer & Blasey, 2004). Specific values as shown in Table 3.

Table 3 and Figure 5 show the influence of intermediary performance on the relationship between alliance conditions, and alliance performance has a moderating role. Therefore, H3 was supported. Also, as seen in Table 4 and Figure 6, the influence of intermediary performance on the relationship between person-organization fit and alliance performance has a moderating role. Therefore, H4 was supported. These results suggest that the relationship between alliance conditions and alliances performance and the relationship between P-O fit and alliances performance are moderated by intermediary performance.

3. Minor questions;

3.1 In the abstract appears p-o fit but the significance of this abbreviation appears in the page.

Response: Thanks for your valuable comment. We have revised it to your comments. 

3.2 Maybe be in the title appears too many times performance.

Response: Thanks for your valuable comment. We revised it per your comments. We changed the title as follows, reducing the word performance.

The Impacts of Conditions and Peron-Organization Fit on Alliances Performance: And the Moderating Role of Intermediary

3.3 The authors must include several references about papers that analyses the moderating effects in the travel industries or others.

Response: Thanks for your valuable comment. We added some references in section 2.3 as follows,

There are many studies work about moderators in strategic alliances of travel industry and others. For instance, Tsaur and Wang (2011) found that competitive intensity moderates the relationship between strategic alliance and performance. Moghaddam et al. (2016) observed SEFs benefit from strategic alliances more if they engage in a moderate number of alliances rather than being overwhelmed with a great number of alliances. Wang, Chin, and Ting (2022) indicated structural capital positively and significantly moderates the mediating effect on the relationship between complementary capabilities and supplier performance. Overall,…

References

1.Moghaddam K, Bosse DA, Provance M. Strategic alliances of entrepreneurial firms: Value enhancing then value destroying. Strategic Entrepreneurship Journal. 2016; 10(2): 153–168.

2.Tsaur SH, Wang CH. Personal ties, reciprocity, competitive intensity and performance of the strategic alliances in Taiwan’s travel industry. The Service Industries Journal. 2011; 31(6): 911–928.

3.Wang C, Chin T, Ting CT. Supply chain capital and firm performance: the role of complementary capabilities. Journal of Intellectual Capital. 2022; https://doi.org/10.1108/JIC-06-2021-0148

---

## [Decision Letter · Decision Letter 1]

26 Sep 2022

The Impacts of Conditions and Person-Organization Fit on Alliances Performance: And the Moderating Role of Intermediary

PONE-D-22-08099R1

Dear Dr. You-Yu Dai,

We’re pleased to inform you that your manuscript has been judged scientifically suitable for publication and will be formally accepted for publication once it meets all outstanding technical requirements.

Kind regards,

María del Carmen Valls Martínez, Ph.D.

Academic Editor

PLOS ONE

Reviewers' comments:

Reviewer's Responses to Questions

**Comments to the Author**

1. If the authors have adequately addressed your comments raised in a previous round of review and you feel that this manuscript is now acceptable for publication, you may indicate that here to bypass the “Comments to the Author” section, enter your conflict of interest statement in the “Confidential to Editor” section, and submit your "Accept" recommendation.

Reviewer #1: All comments have been addressed

Reviewer #3: All comments have been addressed

2. Is the manuscript technically sound, and do the data support the conclusions?

Reviewer #1: Yes

Reviewer #3: Yes

3. Has the statistical analysis been performed appropriately and rigorously? 

Reviewer #1: Yes

Reviewer #3: Yes

4. Have the authors made all data underlying the findings in their manuscript fully available?

Reviewer #1: (No Response)

Reviewer #3: Yes

5. Is the manuscript presented in an intelligible fashion and written in standard English?

Reviewer #1: Yes

Reviewer #3: Yes

6. Review Comments to the Author

Reviewer #1: Dear authors

You have adequately addressed the comments raised in a previous round of review and I feel that this manuscript is now acceptable for publication. Best Regards

Reviewer #3: (No Response)

7. PLOS authors have the option to publish the peer review history of their article (what does this mean?). If published, this will include your full peer review and any attached files.

Reviewer #1: No

Reviewer #3: No

---

## [Editor Report · Acceptance letter]

18 Oct 2022

PONE-D-22-08099R1 

The Impacts of Conditions and Person-Organization Fit on Alliances Performance: And the Moderating Role of Intermediary 

Dear Dr. Dai:

I'm pleased to inform you that your manuscript has been deemed suitable for publication in PLOS ONE. Congratulations! Your manuscript is now with our production department. 

Kind regards, 

on behalf of

Dr. María del Carmen Valls Martínez 

Academic Editor

PLOS ONE